# Continuous Time Evidential Distributions for Irregular Time Series

**Taylor W. Killian** [1 2 *]   **Haoran Zhang** [2]   **Thomas Hartvigsen** [2]   **Ava P. Amini** [3]

## Abstract

Prevalent in many real-world settings such as healthcare, irregular time series are challenging to formulate predictions from. It is difficult to infer the value of a feature at any given time when observations are sporadic, as it could take on a range of values depending on when it was last observed. To characterize this uncertainty we present EDICT, a strategy that learns an evidential distribution over irregular time series in continuous time. This distribution enables well-calibrated and flexible inference of partially observed features at any time of interest, while expanding uncertainty temporally for sparse, irregular observations. We demonstrate that EDICT attains competitive performance on challenging time series classification tasks and enabling uncertainty-guided inference when encountering noisy data.

## 1. Introduction

Irregularly-sampled, multivariate time series data are common across many real-world applications, including healthcare (Jensen et al., 2014), meteorology (Shi et al., 2015), and business (Batres-Estrada, 2015). These data are characterized by missing and/or irregular observations, inducing inherent uncertainty about the value of unobserved features. Traditional machine learning (ML) approaches manage this irregularity through imputation and resampling of the data to impose full observations recorded at evenly spaced time steps. However, the choice of resampling rate and how to fill missing intervals can introduce unnecessary bias in the data used to train models, limiting their reliability. To build reliable predictive models from irregular data it is crucial to model the distribution of the time series—conditioned on sparse previous observations—to infer possible values of unobserved features and allow the natural presentation of the data to dictate what the model learns. Additionally,

precise uncertainty estimates can help interpret predictive confidence and assess the robustness of model predictions.

Prior approaches that attempt to quantify uncertainty over irregular time series either stabilize a learned latent representation of the data (De Brouwer et al., 2019; Fortuin et al., 2020) or forecast future values (Stankeviciute et al., 2021; De Brouwer et al., 2022; Sun and Yu, 2022) as a means to identify possible outcomes of the current observed state. However, these approaches either do not connect the estimated representations with the reliability of downstream predictions (De Brouwer et al., 2019; Sun and Yu, 2022), or fail to provide continuous-time projections of missing values in the time series (Fortuin et al., 2020; Stankeviciute et al., 2021).

Rather than relying on indirect estimates of the predictive distribution (Gal and Ghahramani, 2016b), efforts with a recently introduced approach called evidential learning aim to model the variance in predictive distributions directly, formulated as an evidence acquisition process (Sensoy et al., 2018; Malinin and Gales, 2018; Amini et al., 2020; Charpentier et al., 2020). These methods provide scalable and calibrated uncertainty estimates but have only been developed for static, univariate supervised classification (Sensoy et al., 2018) and regression (Amini et al., 2020; Meinert and Lavin, 2021) problems. The extension of these techniques to irregularly sampled time series presents nuanced complexities due to the sporadic observations, high rates of missingness, and data generation occurring through a dynamic, continuous-time process.

In this work, we present a method to process multivariate irregular time series by through a neural ODE (Kidger, 2021) to construct a latent representation that is used to predict the parameters of a higher-order probability distribution (Figure 1). Under this paradigm, the training data is assumed to provide evidence for the predicted distribution and estimated uncertainties are associated with erroneous model behavior or out of distribution input data. This *evidential distribution* provides calibrated, temporally correlated uncertainty estimates over the multivariate time series and enables robust, uncertainty-guided classification using the learned latent representation. In total, this work makes the following contributions:

---
*Work performed during an internship at Microsoft Research. [1]University of Toronto, Vector Institute [2]Massachusetts Institute of Technology [3]Microsoft Research. Correspondence to: Taylor Killian <twkillian@cs.toronto.edu>.

*Workshop on Interpretable ML in Healthcare at International Conference on Machine Learning (ICML)*, Honolulu, Hawaii, USA. 2023. Copyright 2023 by the author(s).

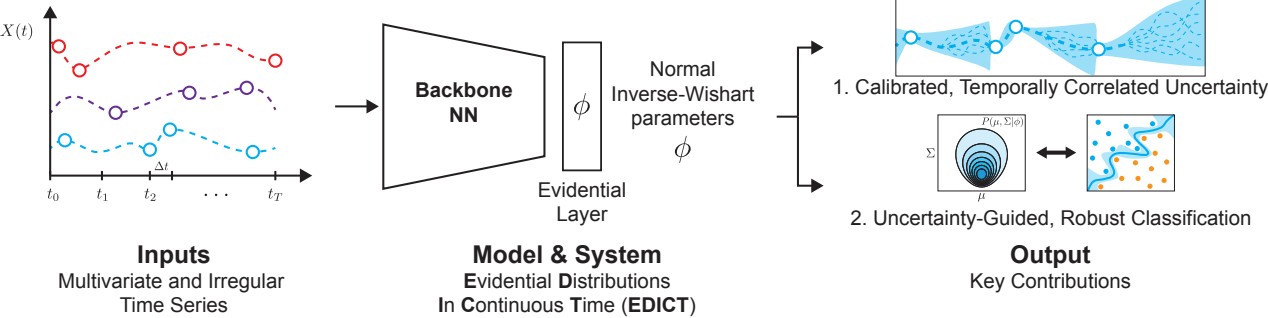

Figure 1: **Learning and deploying Evidential Distributions In Continuous Time (EDICT).** Given a multivariate time series, a continuous-time model is trained to predict the parameters of an evidential distribution that enables calibrated and temporally correlated uncertainty estimation.

1) Formulation of an evidential learning approach for sequential, continuous-time problems;

2) A scalable method for inferring Evidential Distributions in Continuous Time (EDICT), deriving uncertainty estimates without the need for sampling during inference, Monte Carlo estimation, or training on out-of-distribution data;

3) Evaluation of the inferred evidential distribution on both interpolation and extrapolation settings for irregular time series, demonstrating superior accuracy and calibration;

4) Demonstration of uncertainty-guided inference in classification settings, correcting for noisy observations to stabilize predictions and improve performance.

## 2. Uncertainty Estimation with Evidential Learning

Calibrated estimates of neural network (NN) uncertainty are important for understanding how well the model will perform on unseen data, derive measures of confidence, as well as providing feedback on data missingness and irregularity. Bayesian methods have been used to characterize model (e.g., epistemic) uncertainty by placing probabilistic priors on NN parameters, using variational approximations and Monte-Carlo sampling to estimate output variance (Kendall and Gal, 2017). However, contemporary NNs are often too large and complex for these methods to be feasible, with additional challenges in choosing the right approximation and priors on the parameters (Yao et al., 2019).

Rather than placing priors on network weights as with Bayesian NNs, recent approaches place priors directly over a Gaussian likelihood function of the network's predictions by framing model learning as an evidence acquisition process (Sensoy et al., 2018; Malinin and Gales, 2018; Amini et al., 2020; Meinert and Lavin, 2021), motivated by Dempster-Shafer theory (Shafer, 1976). In these approaches, the NN produces the hyperparameters of the posterior distribution via maximum likelihood estimation (Charpentier et al., 2020), where the support of the distribution is derived directly from the training data. Thus, evidential learning provides a scalable, well-calibrated uncertainty estimation technique by eliminating the need for sampling-based or variational approaches.

In this paper, we present a multivariate formulation of evidential distributions for continuous-time NN models, termed Evidential Distributions in Continuous Time (EDICT). EDICT enables prediction of feature evolution as well as calibrated model uncertainties over sequential observations made at irregular time intervals. To our knowledge, EDICT is the first method to develop evidential distributions with corresponding measures of uncertainty over high-dimensional features sets as well as in sequential settings.

## 3. Evidential Distributions in Continuous Time

In this section we formalize our problem statement and outline the EDICT method, specifically focused on developing estimates of NN uncertainty for irregular time series. This is done without the need for sampling during inference (Fortuin et al., 2020; Kingma et al., 2015), Monte Carlo estimation (Gal and Ghahramani, 2016b;a), or training on out-of-distribution data (Hafner et al., 2020; Malinin and Gales, 2018; Malinin et al., 2020) which facilitates far more scalable uncertainty quantification.

### 3.1. Problem setup

We consider a $D$-dimensional irregular time series $X(t)$ with finite observation horizon $T$ and infer a distribution over the feature space, conditioned on previous observations, to forecast the possible values of any feature before it is observed again. We assume that each time series $X(t)$ is measured at $K$ time points $\boldsymbol{t} \in \mathbb{R}^K$ that may arise sporadically, with some subset of the $D$ features present at each observation. Additionally, we assume that a representation $h(t)$ of the latent generating process of the observable time series $X_i(t)$ can be inferred in continuous time. After time $T$ has been reached, and the allowed observation window has closed, we expect to formulate a prediction about some target characteristic $y$ of the time series.

For convenience in modeling, we assume that $X(t)$ is drawn from a multivariate Normal distribution with mean $\mu(t) \in \mathbb{R}^D$ and covariance $\Sigma(t) \in \mathbb{R}^{D \times D}$ such that $X(t) \sim \mathcal{N}(\mu(t), \Sigma(t))$[1]. We aim to model these unknown functions by predicting hyperparameters of their generating distribution using the learned continuous-time representation $h(t)$ formed from prior observations of $X(t)$. Specifically, we place priors on $\mu$ and $\Sigma$ to flexibly account for the possible variance among features of the observed time series $X(t)$, developing an evidential distribution in continuous time. Next we describe:

i) How $h(t)$ is updated in continuous time with neural ODEs;

ii) How the evidential learning objective is used to formulate continuous-time distributions;

iii) How the resulting Evidential Distribution in Continuous Time (EDICT) can be used to guide the reweighting of noisy and out-of-distribution observations;

iv) How $h(t)$ inferred from EDICT can then be used to train downstream classification models to predict underlying characteristics or target values $y$ of the time series $X(t)$.

### 3.2. Continuous-time Representations of Irregular Time Series

Our evidential learning objective maximizes the likelihood of observing $X(t)$, given the parameters $\phi$ of the generating distribution and the prior observations $X(l), \forall\, l < t$:

$$\max_{\phi} p(X(t) \mid \phi, X(l)_{l<t}).$$

We resolve conditional dependence on $X(l)_{l<t}$ by encoding the observations in a continuous-time, recurrent latent representation $h(t)$. Continuous-time (CT) models for sequential processing of information were originally proposed as an extension of recurrent NNs (Funahashi and Nakamura, 1993; Chow et al., 2000). Recently, deep learning models utilizing differential equation solvers have extended these ideas (Chen et al., 2018; Rubanova et al., 2019; Kidger, 2021), enabling a flexible basis for learning the latent dynamics of temporal observations, including those that are irregularly sampled. CT models form a latent representation $h(t)$ of the dynamics underlying the data generating process that is propagated between recorded observations.

To update $h(t)$ once an observation has been made, we employ a GRU-inspired procedure (De Brouwer et al., 2019) that (1) propagates $h(t)$ between observations using a neural ODE $f_{\text{ODE}}$, and (2) adjusts $h(t)$ with an approximate Bayesian mechanism, $f_{\text{Bayes}}$, once an observation $X(t[k])$

---

[1]Time dependence of the parameters of the generating distribution is assumed through the remainder of this paper unless necessary to establish context.

is made. Both of these mechanisms are represented as NNs. To produce the parameters $\phi$ of the generating distribution a third NN, $f_{\mathcal{N}}(\cdot)$ is used to map $h(t)$ to $\phi$. If needed, an additional function $f_{\text{enc}}(\cdot)$ may be used to process the set of observed features $X(t[k])$ to improve the update to $h(t)$. Together, the process of propagating and updating the representation $h(t)$ given observations $X(t[k])$ is as follows:

1. *Propagate $h(t)$ over the interval between observations:*
   $h(t_-) = f_{\text{ODE}}(h(t), t[k] - t[k-1])$

2. *Update $h(t)$ once an observation is made:*
   $h(t_+) = f_{\text{Bayes}}(h(t_-), f_{\text{enc}}(X(t[k])))$

3. *Produce $\phi$ to infer the distribution over $X(t)$ via:*
   $\phi = f_{\mathcal{N}}(h(t)),$

where $t_-$ and $t_+$ denote the time directly before and after $t[k]$. We use this procedure to create a CT representation of the irregular time series. The resulting representation is then used to produce the hyperparameters of an evidential distribution, as outlined in the next subsection.

### 3.3. Constructing EDICT

**Estimating parameters of the evidential distribution** We assume the time series $X(t)$ is drawn from a multivariate Normal distribution with mean $\mu$ and covariance $\Sigma$: $X(t) \sim \mathcal{N}(\mu, \Sigma)$. We formulate the evidential distribution by placing priors on the parameters $\mu$ and $\Sigma$ using a conjugate prior, the Normal Inverse Wishart (NIW) distribution. This entails placing a Gaussian prior on $\mu$ and an Inverse-Wishart prior on $\Sigma$:

$$(\mu, \Sigma) \sim \text{NIW}_{\phi}(\mu, \Sigma | \mu_0, \lambda, \Psi, \nu)$$

where $\quad \mu \sim \mathcal{N}(\mu | \mu_0, \lambda^{-1}\Sigma)$ and $\Sigma \sim \mathcal{W}^{-1}(\Sigma | \Psi, \nu)$.

The parameters $\phi = \{\mu_0, \lambda, \Psi, \nu\}$ of this evidential conjugate prior can be interpreted in terms of "virtual observations" (Jordan, 2009) in support of the evidence collected via the training data. In our evidential learning framework, the parameters $\phi$ are produced by a learned function $f_{\text{NIW}}(\cdot)$ of the CT representation $h(t)$. Thus, any set of features included with the next observation $X(t[k])$ will improve the posterior estimates of the evidential distribution. The parameters $\phi$ of the evidential distribution are inferred and updated through time to approximate the full time series $X(t)$.

Within this evidential framework, the objective is to maximize the likelihood of observing $X(t)$ given $\phi$ and prior observations $X(l), \forall\, l < t$. We formulate this using the

evidential NIW prior as:

$$p(X(t) \mid \phi, X(l)_{l<t}) =$$

$$\iint_{\mu, \Sigma} p(X(t) \mid \mu, \Sigma) \, \mathrm{NIW}(\mu, \Sigma \mid \phi, X(l)_{l<t})$$

$$= \mathcal{T}_{\mathrm{st}} \left( \mu_0, \frac{1+\lambda}{\lambda(\nu - D + 1)} \Psi, \nu - D + 1 \,\Big|\, X(l)_{l<t} \right)$$

(1)

where $\mathcal{T}_{\mathrm{st}}$ denotes the multivariate, $D$-dimensional $t$-distribution (Meinert and Lavin, 2021)[2]. By predicting the parameters $\phi$ using the prior observations $X(l)_{l<t}$ to encode and propagate $h(t)$, we directly estimate the likelihood $p(X(t) \mid \phi, X(l)_{l<t})$ without the need for sampling or other complex inference techniques.

**Using inferred NIW parameters to make predictions** By learning a CT representation (Section 3.2), producing $h(t)$, and learning the NIW evidential distribution, we can flexibly predict the values of $X(t)$ at any time before the observation horizon $T$. With predicted parameters $\phi = \{\mu_0, \lambda, \Psi, \nu\}$ of the NIW distribution, predictions and associated measures of uncertainty of missing features can be made directly. These measures of uncertainty include both irreducible stochasticity in the data distribution (i.e., the aleatoric uncertainty) and the accuracy of the model (i.e., the epistemic uncertainty):

$$\mathrm{Prediction}: \mathbb{E}[\mu] = \mu_0 \quad \mathrm{Aleatoric}: \mathbb{E}[\Sigma] = \frac{\Psi}{\nu - D - 1}$$

$$\mathrm{Epistemic}: \mathrm{var}[\mu] = \frac{\Psi}{\lambda(\nu - D - 1)}$$

The epistemic uncertainty is determined by the variance of the predictions formulated through the NIW. We provide a demonstration of EDICT in Figure 2 using a 2D synthetic dataset of anti-correlated periodic signals with decaying amplitude and random phase shifts. Here, EDICT quickly resolves the correct dynamics of the signals after a small number of observations. EDICT smoothly propagates the predicted uncertainty over intervals where no observations are made, and appropriately contracts this estimate once an observation is made.

**Optimizing NIW parameters** All modules that contribute to the formation of EDICT (CT representation learning and prediction of $\phi$) are optimized as a multi-task learning problem. First, we seek to maximize the model evidence according to the observations of $X(t)$. We do this by minimizing the negative log-likelihood of the multivariate $t$-distribution (Eqt. 1; $\mathcal{L}^{NLL}$). Second, we want to promote that the inferred NIW distribution covers the empirical distribution of the recorded observations through a Kullback-Leibler constraint ($\mathcal{L}^{KL}$). This forces the model to mimic a

---

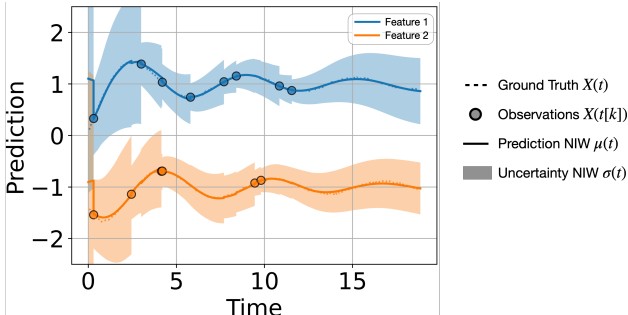

Figure 2: **Evidential uncertainty estimation in irregular time series.** EDICT accurately infers the distribution over a 2D irregular time series, with temporal propagation of uncertainty over intervals of time between observations.

Bayesian update (De Brouwer et al., 2019) when observations are made. Finally, the learned evidential distribution is further regularized by reducing the inferred evidence when the prediction is wrong ($\mathcal{L}^R$) (Amini et al., 2020). Altogether our total objective for learning the continuous-time evidential distribution is:

$$\mathcal{L} = \mathcal{L}^{NLL} + \beta_1 \mathcal{L}^{KL} + \beta_2 \mathcal{L}^R$$

(2)

where $\beta_1$ and $\beta_2$ are hyperparameters that modify the effect of the two regularization terms. Details on this objective (Section B) and on EDICT and its training (Section D) are in the Appendix.

### 3.4. Downstream prediction with EDICT

With a well-calibrated evidential distribution in continuous time, EDICT enables two key abilities in formulating downstream predictions for a target $y$ given an input time series $X(t)$.

First, EDICT learns a rich representation of the latent dynamics of the time series, $h(t)$, that is influenced by the inferred evidential distribution and corresponding estimates of uncertainty. Using this representation $h(t)$, we can develop expressive prediction models $f_{\mathrm{CLF}}(h(t))$ of the target label $y$. We demonstrate this empirically through classification experiments using various irregular time series datasets (Section 4), with results in Section 5.2.

Second, EDICT learns an underlying distribution that provides an explicit measure of the likelihood of any future observation. Consider a deployment scenario where observed features begin to shift out of distribution or become overly noisy through time, common in persistent sensing settings. When constructing a CT representation, the predicted mean of the distribution "jumps" to cover these noisy observations, resulting in an unstable representation $h(t)$. To address this, we design an algorithm that uses EDICT to infer how reliable an observation is and "correct" it, if needed, using the predicted mean and variance as a form of reweighting. This stabilizes the update of the CT representation and thereby

---

[2]See Appendix C.2 and C.3 of Meinert and Lavin (2021)

the predicted mean of the distribution. We designate this as Evidential Distribution Guided Reweighting (EDGR) (Algorithm 1). Using EDGR within EDICT allows for practical use of the learned distribution in downstream classification to provide uncertainty-guided predictions.

---

**Algorithm 1** Evidential Distribution Guided Reweighting

1: **EDGR**(EDICT, T, $\eta$, $X_i(t)$, $h_i(t)$)
2: **for** $k = 1, \ldots, K_i$ **do**
3:      // ODE evolution to $t[k]$
4:      $h(t_-) = f_{\text{ODE}}(h(t), t[k] - t[k-1])$
5:      // Predict NIW parameters, compute $\hat{\mu}$, $\hat{\sigma}$
6:      (Section 3.3)
7:      // Check whether observation is OOD. If so, reweight
8:      **if** $\|X(t[k]) - \hat{\mu}\| > \eta \cdot \hat{\sigma}$ **then**
9:          $X(t[k]) = \text{clip}(\hat{\mu} - \eta \cdot \hat{\sigma}, \ \hat{\mu} + \eta \cdot \hat{\sigma})$
10:     **end if**
11:     // Update $h(t)$ with the observation $X(t[k])$
12:     $h(t_+) = f_{\text{Bayes}}(h(t_-), f_{\text{enc}}(X(t[k])))$
13: **end for**
14: // ODE evolution to T
15: $h(T) = f_{\text{ODE}}(h(t), T - t[K])$
16: // Formulate a prediction for $X(t)$
17: $\hat{y} = f_{\text{CLF}}(h(T))$

---

EDGR establishes whether an observation $X(t[k])$ is noisy enough to require reweighting if it deviates more than $\eta * \hat{\sigma}$ from the predicted mean $\hat{\mu}$, where $\eta$ is a hyperparameter to balance the stability of EDICT with reductions in the contributions of noisy data. We show the utility of EDGR through experiments with additive heteroskedastic noise of increasing magnitude through time (Section 5.2).

# 4. Experiments

## 4.1. Datasets

We evaluate EDICT on one synthetic and four publicly available irregular time series datasets (Table 1):

1. `Synthetic`: We sparsely sample 3 periodic features, forming a binary prediction task. The outcome $y$ is informed by feature 1 or 2 with the third feature being uninformative.

2. `Gestures`: The uWave dataset (Liu et al., 2009) contains recordings of a three-axis accelerometer used to "draw" 8 pre-defined templates. We subsample the data to keep a random 10% of all observations.

3. `Activity`: The Human Activity dataset (Kaluža et al., 2010) contains recordings of subjects performing various physical activities, consisting of the 3D position of 4 monitors (12 features in total).

4. `PhysioNet`: The goal is to predict in-hospital mortality for patients in the ICU. The task is provided by the PhysioNet/Computing in Cardiology Challenge 2012 (Silva et al., 2012).

5. `MIMICMort`: The task is to predict in-ICU mortality from labs and vitals. We use MIMIC-Extract (Wang et al., 2020), which is derived from the MIMIC-III Clinical Database (Johnson et al., 2016).

Table 1: Datasets used for empirical evaluations.

| Dataset | # Classes | # Features | # Samples |
|---|---|---|---|
| Synthetic | 2 | 3 | 10,000 |
| Gestures | 8 | 3 | 4,478 |
| Activity | 7 | 12 | 10,486 |
| PhysioNet | 2 | 34 | 7,990 |
| MIMICMort | 2 | 49 | 24,391 |

Dataset details, including preprocessing procedures, are contained in the Section C in the Appendix.

## 4.2. Baselines

For baseline comparison, we focus on methods that develop measures of uncertainty based on the continuous-time arrival of irregular observations. Our primary baseline is GRU-ODE-Bayes (De Brouwer et al., 2019), which uses a GRU-based ODE to evolve the hidden state continuously between observations, while using a Bayesian update to transform the hidden state at each observation. To our knowledge, GRU-ODE-Bayes is the only method, prior to our present work, that can provide predicted distributions for continuous-time observations in multivariate, irregularly-sampled time series.

To evaluate downstream classification, we also compare against contemporary NN methods that handle irregular time series for classification alone: (i) Interpolation Prediction Network (IPN) (Shukla and Marlin, 2019), (ii) GRU-D (Che et al., 2018), (iii) Set Functions for Time Series (SeFT) (Horn et al., 2020). These methods do not produce uncertainty estimates as EDICT does, but are included for completeness.

## 4.3. Experimental Setup

All datasets are split into training, validation, and evaluation subsets following a 70/10/20 split, stratified such that the proportion of class labels is consistent among all subsets. The same data splits are used when training EDICT and on the downstream classification tasks. Results in Section 5 are derived from the evaluation subset. We perform three types of experiments to evaluate EDICT.

### 4.3.1. UNCERTAINTY EVALUATION AND CALIBRATION

As our primary empirical evaluation, we train EDICT models on each dataset using the loss shown in Equation 2, varying $\beta_1$ and $\beta_2$ among other hyperparameters (see Appendix, Section D). Note that this procedure is unsupervised, i.e., it does not make use of exogenously provided task labels but

rather utilizes the sequentially gathered feature observations themselves. We evaluate the quality of the predictions made by EDICT and their accompanying uncertainty measures via its evidential distribution, compared against GRU-ODE-Bayes and its inferred distribution. We assess the following metrics (Sun and Yu, 2022):

1. **Coverage**. We want inferred model uncertainties to be well *calibrated*. For $\alpha \in [0, 0.5]$, we select the $1 - 2\alpha$ confidence region of the distribution and compute the fraction of observed data within it. We quantify this measure of coverage with Expected Calibration Error (ECE) (details in Appendix, Section D).

2. **Efficiency**. It is preferable for a calibrated model to have small *area* (i.e., the width of the confidence intervals). We plot the average width of the confidence region over all $\alpha$.

3. **Forecasting error**. We evaluate how far the mean of the predictive distribution is from the actual observation by computing the Mean Squared Error (MSE).

We evaluate these metrics in both *interpolation* (Section D) and *extrapolation* (Figure 3, Table 2) settings. We hold out a random 10% of observations prior to time $t_{\text{cut}}$. During training, we only expose the model to the remaining data in $[0, t_{cut})$. We evaluate *interpolation* by evaluating with the held-out data. For *extrapolation*, we use unobserved data from $[t_{cut}, T]$. Each metric is calculated separately for each time series, and a confidence interval over each metric is generated as one standard deviation over the samples in the dataset.

### 4.3.2. PERFORMANCE IN DOWNSTREAM DECISION-MAKING

Next, we evaluate EDICT's performance in downstream classification tasks. We freeze the weights of the EDICT model, and train a downstream linear classifier $f_{\text{CLF}}$ using the embedded latent representations $h(t)$ of $X(t)$. We compute the accuracy of the predictions, and generate confidence intervals by repeating this procedure with three random seeds.

### 4.3.3. PREDICTION OVER NOISY OBSERVATIONS

To evaluate our method in a realistic scenario where the test samples may be out of distribution from the training data (e.g., due to sensor failure), we experiment with injecting temporally compounding heteroskedastic noise into the test-set observations with the degree of variance being a function of time. For each dataset, a range of 10 noise levels are chosen as the base rate for this exponentially compounding amount of noise (details in the Section D).

We demonstrate the utility of EDICT for uncertainty-guided classification by using EDGR (Section 3.4, Alg. 1) to reweight observations identified as outliers. We compare the performance of this method on $f_{\text{CLF}}$ AUROC against base EDICT (i.e., no reweighting), as well as against an alternative approach which clips observations based on the data distribution's population mean. In all experiments, we use $\eta = 1.96$, corresponding to a 95% confidence interval.

## 5. Results

We first evaluate the quality of EDICT's uncertainty bounds over feature values and show that EDICT achieves better calibration than GRU-ODE-Bayes on several datasets (Section 5.1). EDICT's performance in downstream classification is then demonstrated to match current state of the art baselines while also providing calibrated uncertainty metrics (Section 5.2). Finally, we illustrate the flexibility of EDICT's uncertainty intervals in accounting for noisy observations. We show that EDICT with EDGR outperforms the the comparative baselines at high noise levels, particularly in complex, real-world datasets (Section 5.3).

### 5.1. Evidential Distribution Calibration

In Figure 3 and Table 2, we compare the calibration of the distributions predicted by EDICT and GRU-ODE-Bayes when extrapolating beyond the observed data for the `Synthetic` and `MIMICMort` datasets (comparisons for all datasets, including interpolation results, can be found in the Appendix, Section D). We find that EDICT has much lower MSE than GRU-ODE-Bayes on `Synthetic` and `Gestures`, and the MSEs are comparable on the remaining three datasets when confidence intervals are taken into account (Table 2). On the `Synthetic` dataset, EDICT achieves greatly improved efficiency relative to GRU-ODE-Bayes and exhibits strong calibration performance relative to both the ideal and GRU-ODE-Bayes baseline coverage (Figure 3A) in all other datasets. On the real-world `MIMICMort` dataset, EDICT demonstrates strong coverage, indicating well-calibrated model uncertainties (Figure 3B).

Table 2: **Extrapolation performance and calibration** of EDICT and GRU-ODE-Bayes.

| | EDICT | | GRU-ODE-Bayes | |
| --- | --- | --- | --- | --- |
| **Dataset** | MSE | ECE | MSE | ECE |
| Synthetic | $0.009 \pm 0.01$ | $0.129 \pm 0.08$ | $0.696 \pm 0.20$ | $0.093 \pm 0.78$ |
| Gestures | $0.273 \pm 0.02$ | $0.066 \pm 0.04$ | $0.603 \pm 0.12$ | $0.054 \pm 0.05$ |
| Activity | $1.126 \pm 0.72$ | $0.127 \pm 0.07$ | $1.119 \pm 0.76$ | $0.085 \pm 0.05$ |
| PhysioNet | $1.326 \pm 1.46$ | $0.008 \pm 0.07$ | $1.160 \pm 0.76$ | $0.014 \pm 0.01$ |
| MIMICMort | $1.374 \pm 2.68$ | $0.066 \pm 0.04$ | $1.227 \pm 0.94$ | $0.009 \pm 0.01$ |

### 5.2. Downstream Classification

One major goal of learning effective latent representations for irregularly sampled time series is to achieve strong performance on some downstream task. To this end, we compare the test-set AUROC of EDICT with a set of baselines in classifying irregular time series (Figure 4; Appendix, Section D.3). On the zero-noise classification task (train and test

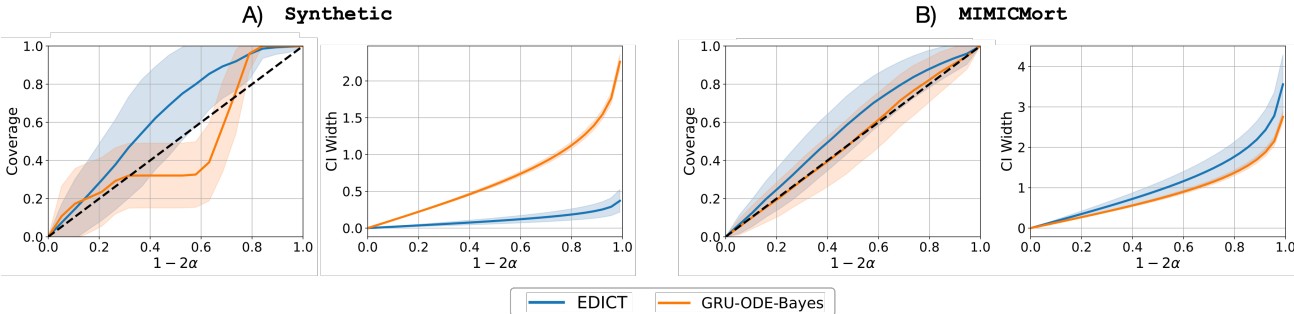

Figure 3: **Calibration of EDICT on the extrapolation task.** Comparison of the extrapolation calibration of the distributions predicted by EDICT and GRU-ODE-Bayes, in terms of coverage and efficiency, for the **(A)** `Synthetic` and **(B)** `MIMICMort` datasets. Ideal coverage would mirror the black dotted line while lower CI width indicates more efficient distributions. Both methods have comparable performance on `MIMICMort`, while EDICT has much better efficiency on `Synthetic`.

samples drawn from the same distribution), EDICT matches the performance of current standard approaches (Figure 4, noise level 0). Critically, the majority of evaluated baselines fail to provide uncertainty intervals over intermediate noise values. Variations of EDICT and GRU-ODE-Bayes are the only methods in Figure 4 that do so, indicating EDICT's ability to enable both strong downstream decision-making and calibrated uncertainties. In high noise settings, all baselines have reduced performance where prediction quality is near chance. Across all methods tested, EDICT is more resistant to performance decay caused by noisy observations. The performance demonstrated in Figure 4 suggests that the latent $h(t)$ learned by EDICT and directly influenced by its evidential distribution – is more robust to these outlying, noisy observations.

### 5.3. Correcting for Noisy Observations

One main advantage of EDICT over most baselines is the ability to infer a distribution over feature values in continuous time. Here, we demonstrate one utility of this distribution, by using it to clip values identified as outliers in the presence of noisy data not seen during training. To this end, we designed an algorithm, EDGR, that leverages the learned evidential distribution to identify outliers and reweight them according to the uncertainty of the observed feature. In Figure 4 (and Appendix, Section D.3), we find that EDICT with uncertainty-guided reweighting (EDICT w/ EDGR) allows the model to maintain robust performance for high noise levels, especially for `Synthetic`, `Gestures`, `Physionet`, and `MIMICMort`, where higher noise levels cause all baseline models, including GRU-ODE-Bayes, to fail.

Taking these results together, we observe that EDICT performs best when there are relationships to be inferred among all feature dimensions in continuous time. Of the experiments presented in this work, four of the five datasets have this characteristic, excepting `Activity`. The features within the `Activity` time series are slightly disjoint

from one another and are not time aligned. Due to this, set-based interpolation methods such as SeFT and IPN are better suited for this dataset. Among all other datasets, we see that embedding the irregular time series in a continuous-time latent representation $h(t)$ allows us to adequately account for missing features when predicting the evidential distribution. In these settings EDICT enables robust prediction performance and the ability to leverage the distributional estimates to mitigate OOD observations.

## 6. Related Work

**Learning from irregular time series**  Irregular time series contain observations sampled at uneven times from constantly-changing environments, representative of most real-world settings. Such irregularity is challenging for machine learning, especially when multiple variables are observed simultaneously. Many traditional works tackle irregularity through imputation, resampling the values of an irregular series at a set of new, evenly spaced timesteps (Lipton et al., 2016; Zheng et al., 2017; Che et al., 2018; Li and Marlin, 2020; Mozer et al., 2017). While imputation has proven useful in forecasting (Cao et al., 2018) and classification (Che et al., 2018), there are serious drawbacks. For instance, choosing a good resampling rate is crucial (Hartvigsen et al., 2023), yet the optimal rate is often unknown *a priori*. Furthermore, imputation can introduce unnecessary bias, since irregularity is often *natural*. These limitations have recently spurred major efforts on learning continuous-time models of irregular time series.

**Continuous-time models and uncertainty estimation**  Continuous-time methods have been shown to learn the latent dynamics of irregular time series (Morrill et al., 2022; Kidger, 2021; Rubanova et al., 2019; Jia and Benson, 2019; Hasani et al., 2021; Schirmer et al., 2022; Salvi et al., 2022; Hasani et al., 2022). By providing access to representations at any desired time, these models are more flexible than their imputation-based precursors and are now the state-of-

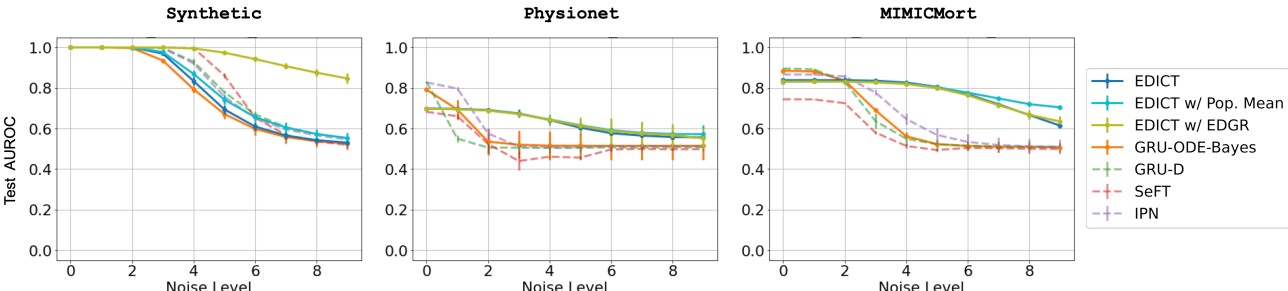

Figure 4: **Classification performance in the presence of noisy observations.** Test AUROC of each method on three binary classification tasks at varying noise levels. We find that all variations of EDICT outperform the GRU-ODE-Bayes baseline at high noise levels. Baseline methods denoted with (---) are not capable of generating uncertainty intervals over intermediate feature values. Results for additional datasets and metrics can be found in Appendix D.3.

the-art approach to many problems. Originally grounded in RNNs (Funahashi and Nakamura, 1993; Chow et al., 2000), most recent approaches succeed by parameterizing differential equations with NNs (Kidger, 2021; Rubanova et al., 2019; Jia and Benson, 2019). Despite recent advances, uncertainty quantification for these continuous-time models remains in its infancy (Graf et al., 2021), although there have been successes estimating the distributions of discrete time series (Rasul et al., 2021a;b; Yu et al., 2021). Indeed, some works have included uncertainty for multivariate irregular time series, especially through multi-task Gaussian processes (Cheng et al., 2020; Fortuin et al., 2020; Ghassemi et al., 2015). However, these approaches are notoriously difficult to scale to high dimensions and fall prey to the resampling challenges native to imputation methods. Our work directly addresses these needs by quantifying uncertainty for continuous-time models while maintaining scalability as a byproduct of using standard deep learning frameworks to construct our evidential distributions.

## 7. Discussion

In this paper we introduce Evidential Distributions in Continuous Time (EDICT), a continuous-time formulation of evidential deep learning for sequential and irregular time series problems. EDICT enables temporally correlated estimates of uncertainty over intervals of missing observations and inconsistent measurement patterns, providing stability to latent inference processes of the underlying data. EDICT maintains distributional calibration on a variety of complex datasets, while achieving competitive performance among current state of the art time series classification algorithms. Further, by virtue of the inferred evidential distribution, EDICT enables improved robustness in the presence of noisy observations, avoiding drastic degradation of model performance through uncertainty-guided inference.

In contrast to traditional uncertainty quantification approaches in contemporary deep learning, EDICT forms its distributions without the use of sampling, complex variational approximations, or training on out-of-distribution

data. Further, it encodes the observations of an irregular time series without needing to make *a priori* assumptions about how to best manage intervals of missingness. This directly overcomes the need to construct imputation strategies manually or to discretize the temporal component of the observed data. As such, we propose EDICT as a flexible method for making predictions and inferring calibrated uncertainties from irregular time series.

**Limitations and Future Work**   The flexibility and scalability of our approach holds promise for potential use in challenging real-world scenarios where forecasting the evolution of features is equally as important as making accurate inference of their current values. Our proposed continuous-time evidential distributions enable both objectives and may provide a decision support framework for practical use by domain experts. While promising, there are some distinct limitations of EDICT. We did not test the method against asymmetric noise models when evaluating the performance of EDGR, the adaptive reweighting approach formulated as a consequence of providing continuous-time measures of uncertainty. Because of this, we cannot make comprehensive, general claims about the applicabilty of EDICT and EDGR across all manners of time series data. Future work will evaluate EDICT's robustness against various forms of noise. Additionally, the methods used to propagate and update the latent representation of the time series can result in entanglement of the predicted distributions over the observable features. Future work will investigate whether enforcing a disentanglement objective serves to improve EDICT and its performance on downstream prediction tasks.

**Contributions**   This work originated while TWK interned with AA at MSR New England. Together, they conceived of and developed the research. TWK carried out the data processing, code development, and experimentation. HZ supported the analysis and contributed to medical dataset preprocessing. TH helped refine research directions, advised on experiments, and helped implement baseline algorithms. Led by TWK, all authors contributed to writing the paper.

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

# Appendix

## A. Potential Negative Societal Impacts

There are several potential negative societal impacts of our work. First, we emphasize that, without proper testing, the implementation of EDICT in safety-critical prediction environments is not guaranteed to ensure reliable performance. We have provided an initial proof of concept that reliability and robustness is improved by learning an evidential distribution in continuous time, yet our evaluations along this dimension are limited in scope. Misuse and mis-deployment of our method could result in negative societal impact. Second, we acknowledge that training large machine learning models may result in high power consumption and carbon emissions, which we did not explicitly quantify in this work. Finally, although EDICT provides robust and calibrated uncertainty estimates, misinterpretation of these estimates could lead to incorrect and mis-guided decisions, especially in critical fields such as healthcare, potentially resulting in real-world harm.

## B. Formulation of the continuous-time evidential distribution training objective

In this section, we highlight the formulation of the multi-task objective used to learn the continuous-time evidential distribution with the following components:

1. $\mathcal{L}^{\text{NLL}}$, Normal Inverse Wishart (NIW) posterior negative log likelihood

2. $\mathcal{L}^{\text{KL}}$, Kullback-Liebler divergence between the NIW evidential distribution and the empirical distribution

3. $\mathcal{L}^{\text{R}}$, Evidential regularization

### B.1. NIW Negative Log Likelihood

In this section we expand on Equation 1, showing the formulation and how it is used to derive the NLL objective used to maximize the model's fit of the evidential distribution.

Recall that we assume the time series to be generated by a multivariate normal distribution, parameterized by the mean $\mu$ and covariance $\Sigma$. We place evidential priors on these parameters using the Normal Inverse Wishart (NIW) since it is conjugate with the assumed generating distribution. The parameters of the NIW distribution are $\phi = \{\mu_0, \lambda, \Psi, \nu\}$ and can be interpreted as "virtual observations". That is, $X(t) \sim \mathcal{N}(\mu, \Sigma)$ and

$$(\mu, \Sigma) \sim \text{NIW}_\phi(\mu, \Sigma | \mu_0, \lambda, \Psi, \nu)$$
$$\text{where} \quad \mu \sim \mathcal{N}(\mu|\mu_0, \lambda^{-1}\Sigma) \text{ and } \Sigma \sim \mathcal{W}^{-1}(\Sigma|\Psi, \nu).$$

We are interested in inferring the distribution of possible values a time series may take at time $t$, given the NIW prior and any previous observations. This takes the form:

$$p(X(t) \,|\, \phi, X(l)_{l<t}) = \iint_{\mu, \Sigma} p(X(t) \,|\, \mu, \Sigma) \,\text{NIW}(\mu, \Sigma \,|\, \phi, X(l)_{l<t})$$

Since the NIW distribution is conjugate with the multivariate normal, an analytic solution to this expression exists and takes the form of a multivariate t-distribution (Murphy, 2007).

$$p(X(t) \,|\, \phi, X(l)_{l<t}) = \mathcal{T}_{\text{st}} \left( \mu_0, \frac{1+\lambda}{\lambda(\nu - D + 1)}\Psi, \nu - D + 1 \,\bigg|\, X(l)_{l<t} \right)$$

As mentioned in Section 3, the conditional dependence on the observations of $X(t)$ prior to time $t$ are resolved through the the propagation of the hidden representation $h(t)$ through the GRU-ODE component of the model that is then used to produce the NIW parameters $\phi$ from $f_{\text{NIW}}(h(t))$. We can then derive the negative log likelihood of this t-distribution in order to arrive at our objective. That is,

$$\mathcal{T}_{\text{st}} \left( \mu_0, \frac{1+\lambda}{\lambda(\nu - D + 1)}\Psi, \nu - D + 1 \,\bigg|\, X(l)_{l<t} \right)$$
$$= \frac{\Gamma(\frac{\nu-D+1}{2} + \frac{D}{2})\det(A)^{-1/2}}{\Gamma(\frac{\nu-D+1}{2})(\nu - D + 1)^{D/2}\pi^{D/2}} \cdot$$
$$\left[ 1 + \frac{1}{\nu - D + 1}(X(t) - \mu_0)^\intercal A^{-1}(X(t) - \mu_0) \right]^{-\frac{(\nu-D+1)+D}{2}}$$
$$= \frac{\Gamma(\frac{\nu+1}{2})\det(A)^{-1/2}}{\Gamma(\frac{\nu-D+1}{2})(\nu - D + 1)^{D/2}\pi^{D/2}} \cdot$$
$$\left[ 1 + \frac{1}{\nu - D + 1}(X(t) - \mu_0)^\intercal A^{-1}(X(t) - \mu_0) \right]^{-\frac{\nu+1}{2}}$$

where $A = \frac{\Psi}{\lambda(\nu-D+1)}$. Then, after taking the log, we arrive at

$$= \log\left( \frac{\Gamma(\frac{\nu+1}{2})}{\Gamma(\frac{\nu-D+1}{2})} \right) - \frac{1}{2}\log\left(\det(A)\right) - \frac{D}{2}\left(\log\left(\nu - D + 1\right) + \log\left(\pi\right)\right)$$
$$- \frac{\nu+1}{2}\log\left( \left[ 1 + \frac{1}{\nu - D + 1}(X(t) - \mu_0)^\intercal A^{-1}(X(t) - \mu_0) \right] \right)$$
$$= \log\left( \frac{\Gamma(\frac{\nu+1}{2})}{\Gamma(\frac{\nu-D+1}{2})} \right) - \frac{1}{2}\log(\det(\Psi))$$
$$- \frac{D}{2}\left( \log\left( \frac{1}{\lambda(\nu - D + 1)} \right) + \log(\nu - D + 1) + \log(\pi) \right)$$
$$- \frac{\nu+1}{2}\log\left( \left[ 1 + \lambda(X(t) - \mu_0)^\intercal \Psi^{-1}(X(t) - \mu_0) \right] \right)$$

With some additional manipulation, the final form of the negative log likelihood objective is:

$$\mathcal{L}^{\text{NLL}} = -\log\left(\frac{\Gamma(\frac{\nu+1}{2})}{\Gamma(\frac{\nu-D+1}{2})}\right) + \frac{D}{2}\log\left(\frac{\pi}{\nu}\right) + \frac{1}{2}\log\left(\det(\Psi)\right) + \frac{\nu+1}{2}\log\left(\left[1 + \lambda(X(t)-\mu_0)^\intercal \Psi^{-1}(X(t)-\mu_0)\right]\right)$$

## B.2. KL divergence

The KL divergence component of our objective is a constraint that forces the model to mimic a Bayesian update following De Brouwer et al. (2019). The intention of this constraint is to encourage the evidential distribution after updating $h(t_-)$ to $h(t_+)$ ($p_{\text{post}}$) to remain close to the a Bayesian posterior of the NIW distribution produced by $h(t_-)$ ($p_{\text{pre}}$) and the empirical distribution of the time series ($p_{\text{obs}}$). Thus, the target distribution is:

$$p_{\text{Bayes}} \propto p_{\text{pre}} \cdot p_{\text{obs}}$$

Thus, we have the KL divergence between $p_{\text{post}}$ and $p_{\text{Bayes}}$ as this component of our objective:

$$\mathcal{L}^{\text{KL}} = D_{\text{KL}}\left(p_{\text{Bayes}}||p_{\text{post}}\right)$$

## B.3. Evidential Regularization

Following Amini et al. (2020), we want to minimize the model evidence when the model produces errors in its prediction of the features that have been observed. We design a penalty term that minimizes the "virtual observations" (and thereby expanding the uncertainty) of the evidential distribution when the mean predictions of the distribution are in error. To formulate this penalty we scale the L1 error of the NIW distribution mean and the true observations by the total evidence of the NIW distribution ($\Phi = (\lambda + \nu)$). That is,

$$\mathcal{L}^{\text{R}} = ||\mu_0 - X(t)||_1 \cdot (\lambda + \nu)$$

# C. Data Details

## C.1. Synthetic data generation

`Synthetic`: We generate a synthetic dataset consisting of multivariate time series with 3 periodic features to form a binary prediction task. The outcome y is informed by feature 1 or 2 (depending on class 0 or 1 respectively) with the third feature serving as a distractor, being correlated with the uninformative feature. These features are sampled such that observations are sparse.

Inspired by medical time series, we generate 10,000 batches of periodic signals with randomized starting points and frequencies. The underlying means of the period signals are also class dependant. This dataset was constructed primarily to qualitatively demonstrate the learned evidential distributions as well as develop a clear and controllable test-bed for measuring the effects of noise applied to the data and

how the proposed EDICT w/ EDGR correction would add robustness to the classification performance. A collection of 3 such sequences are presented in Figure 5.

After generation, the signals are made irregular by masking out no less than 75% of the features, with a single requirement that there is a least one feature that has an observation contained in an initial observation window. We then split the generated data into train/val/test subsets, ensuring to stratify by the correlated/not-correlated label. We maintain 70% of the generated data as a training set, 10% for validation, and 20% for the held out test set.

## C.2. Data Processing and Information

### C.2.1. uWave Gestures

`Gestures`: The uWave gesture dataset (Liu et al., 2009) consists of recordings of a three-axis accelerometer within a hand-held device that human subjects used to "draw" simple gesture patterns following pre-defined templates, divided into 8 categories. We sub-sample the dataset to contain a random 10% of all features. This data is provided as a regularly sampled, dense timeseries with observations at 100 Hz for 3.15 seconds. We subsample the dataset to contain no more than 10% of the features in a particular time series. We maintain 70% of the data as a training set, 10% for validation, and 20% for the held out test set. All features were z-normalized based on the mean and standard deviation of the training set.

### C.2.2. Human Activity Dataset

`Activity`: The Human Activity dataset (Kaluža et al., 2010) contains time series of five individuals performing various activities: walking, sitting, lying, etc. The data consists of the 3D position of monitors attached to their belt, chest and ankles (12 features in total). We followed the same procedure as Rubanova et al. (2019) to prepare the data, partitioning the lengthy time series into non-overlapping windows and shifting the observed features so that no more than three features were observed at the same time. We maintain 70% of the data as a training set, 10% for validation, and 20% for the held out test set. All features were z-normalized based on the mean and standard deviation of the training set.

### C.2.3. PhysioNet

`PhysioNet`: The goal is to predict in-hospital mortality for patients in the ICU. The task is provided by the PhysioNet/Computing in cardiology challenge 2012 (Silva et al., 2012). We follow the standard pre-processing protocol,

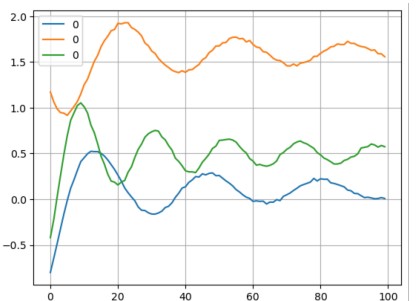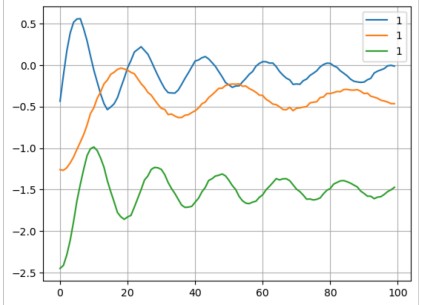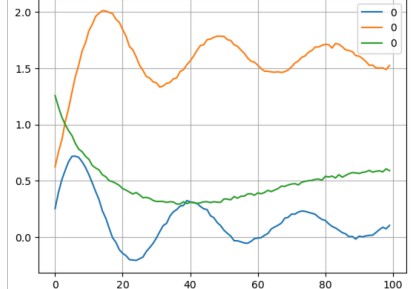

Figure 5: Examples of the synthetic data. Here we demonstrate 3 sequences of 3 dimensions where the features are either informative feature is either feature 1 or 2 depending on the class (0 or 1) while feature 3 serves as a distractor being correlated with the uniformative feature.

removing outliers and ensuring that non-physical measurments are discarded. We further aggregated observations into 10-minute long intervals to reduce the length of the time series for any one patient. We maintain 70% of the data as a training set, 10% for validation, and 20% for the held out test set. All features were z-normalized based on the mean and standard deviation of the training set.

### C.2.4. MIMIC

`MIMICMort`: The task is to predict in-ICU mortality from labs and vitals. We use MIMIC-Extract (Wang et al., 2020), which is derived from the MIMIC-III Clinical Database (Johnson et al., 2016). We downloaded a pre-processed csv of the baseline settings of the MIMIC-Extract tool, directly from the repo. We further processed the data to remove outliers and unify dimensions among features. Additionally, we discarded all features that were more that 95% missing. We maintain 70% of the data as a training set, 10% for validation, and 20% for the held out test set. All features were z-normalized based on the mean and standard deviation of the training set.

## D. Experimental and Model Details

### D.1. Model details

To formulate the continuous-time evidential distribution we leverage a set of connected neural modules. These are:

- $f_{\text{ODE}}$: the continuous-time model that propagates the hidden representation of the time series between observations. Produces $h(t_-)$.

- $f_{\text{Bayes}}$: a recurrent module that is used to update $h(t_-)$ to $h(t_+)$ following the encoding of collected features the time series at time $t$.

- $f_{\text{enc}}$: an encoding function of the observed features $X(t)$.

- $f_{\text{NIW}}$: the function that produces the parameters of the NIW evidential distribution from the hidden representation $h(t)$. Internal to this model are separate

submodules for each parameter of $\phi$. That is, separate small neural networks are used to produce each of $\{\mu_0, \lambda, \Psi, \nu\}$.

The relationship between these modules functionally is as follows:

$$h(t_-) = f_{\text{ODE}}(h(t), t[k] - t[k-1])$$
$$h(t_+) = f_{\text{Bayes}}(h(t_-), f_{\text{enc}}(X(t[k])))$$
$$\phi_{\text{NIW}} = f_{\text{NIW}}(h(t))$$

For convenience and to help ensure that $\Psi$ is positive definite, we only produce the diagonal of the matrix from the corresponding submodule and treat these outputs as the log-variance for each feature dimension. Thus to form $\Psi$ we exponentiate the outputs from its corresponding submodule and then create a diagonal matrix of the resulting vector.

For the experiments presented in this paper we use the following parameter settings.

- Internal to the $f_{\text{ODE}}$ model, there is an initialization layer that encodes static covariates of the time series following the work of De Brouwer et al. (2019). This initializes $h(t)$ with a specified dimension chosen as the hyperparameter. The internal layers of the $f_{\text{ODE}}$ module do not modify the size of this representation.

- The $f_{\text{Bayes}}$ module takes the output of the function $f_{\text{enc}}$ and maps the encoding to the hidden state using a standard GRU cell (Cho et al., 2014).

- $f_{\text{enc}}$ maps the observed features of $X(t)$ to a 25-dimensional encoding.

- All submodules of $f_{\text{NIW}}$ are comprised of small 2-layer neural networks that map $h(t)$ to the parameters of the NIW distribution. The internal hidden layer is set to have 25 dimensions.

- When formulating predictions of whether the features of the time series are correlated or not, we train a separate classifying function $f_{\text{clf}}$ that maps the hidden

representation $h(t)$ to the binary prediction. We kept $f_{\text{clf}}$ small, again utilizing a small 2-layer neural network with the internal hidden layer having half as many dimensions as $h(t)$ to produce a bottleneck layer prior to formulating the final predictions.

## D.2. Experimental details

### D.2.1. PRETRAINING EDICT

The training procedure for EDICT follows directly from GRU-ODE-Bayes (De Brouwer et al., 2019) with some important adjustments to better reflect the inference task for the multivariate evidential distribution. First, we exchanged the linearized univariate version of the KL divergence metric, replacing it with the appropriate Multivariate KL divergence function. We additionally implemented the NIW NLL and Evidential Regularization loss terms as outlined in Section B of this Appendix.

While training EDICT to best infer the evidential distribution, we performed hyperparameter tuning for the following variables:

$\beta_1$ and $\beta_2$ to account for the effect of the regularization terms in the computation of the training objective. The learning rate, training batch size, the number of training epochs, the number of layers and hidden units in each neural network component, and finally the dimension of $h(t)$. We performed this tuning for each dataset, selecting the model that had the lowest interpolation MSE on the validation dataset.

### D.2.2. TRAINING CLASSIFICATION MODELS

We trained linear classification model on top of the latent representation $h(t)$ provided by EDICT as briefly outlined in Section 3.1. We performed hyperparameter tuning for the classification models by varying the learning rate, number of training epochs, and batch size. We also trained separate random initializations using the best hyperparameters with 3 separate seeds. The best performing classification model for each dataset was selected based on validation set accuracy.

The noise applied to the evaluation data for each dataset was generated by choosing from a pre-defined set of base increasing noise rates, specified by the capacity of each dataset to have classifiers have maximized predictive entropy. In order to model the case laid out in the main body of the paper, where the noise compounds over time. We developed a time-dependent noise model that would generate gaussian noise with increasing variance proportional to the time each observation was made. In practice the generated noise was sampled from a zero-mean gaussian with standard deviation ("scale") according to the following relationship:

$$\texttt{scale} = 0.1 * \text{level}^{t[k]}.$$

### D.2.3. EXPECTED CALIBRATION ERROR

To compute the Expected Calibration Error (ECE (Nixon et al., 2019)), we select twenty values of $1 - 2\alpha$ in a grid between 0 and 1. For each value, we compute the fraction of data points covered by the confidence region, subtract $1 - 2\alpha$, and take the absolute value. As each bin has the same number of samples, we compute the ECE by taking the unweighted average.

## D.3. Additional Results

### D.3.1. CALIBRATION OF THE INFERRED DISTRIBUTIONS

In Table 3 we present the full calibration results, for both interpolation and extrapolation predictions made using the distributions inferred using EDICT and GRU-ODE-Bayes (De Brouwer et al., 2019), respectively.

In Figures 6 – 10 we present the calibration comparison between EDICT and GRU-ODE-Bayes for all datasets.

### D.3.2. CLASSIFICATION PERFORMANCE

In Table 4 we present the full classification results for the test accuracy metric, comparing EDICT to several baselines and ablations. In Table 5, we compare all methods for datasets that contain a binary prediction task. We see that as the noise level increases, generally model performance decreases. However when applying reweighting, either with EDGR or using the population mean, that the performance of EDICT recovers and happens to outperform all the baselines on three of the five datasets (Synthetic, MIMIC, Physionet).

In Figures 11– 15 we present all of the classification performance overviews for each dataset.

EDICT performs best when there are relationships to be inferred among all feature dimensions in continuous time. Of the experiments presented in this work, four of the five datasets have this characteristic, excepting `Activity`. By construction, the features within the `Activity` time series are slightly disjoint from one another and are not time aligned (there are four blocks of three features each, corresponding to a 3-axis accelerometer, combined into a time series with some offset after each block). Due to this, set-based interpolation methods such as SeFT and IPN are better suited for this dataset. In other datasets, we see that embedding the irregular time series in a continuous-time latent representation $h(t)$ allows us to adequately account for missing features when predicting the evidential distribution. This enables robust prediction performance and the ability to leverage the distributional estimates to mitigate OOD observations.

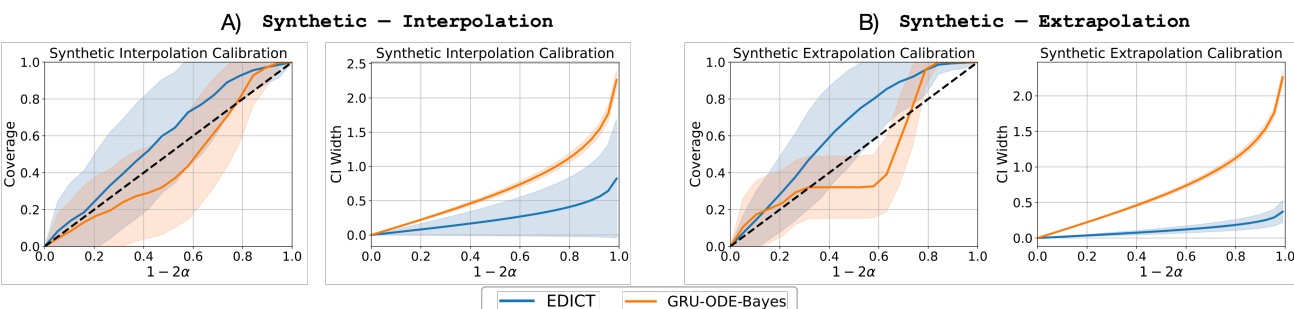

Figure 6: **Calibration of EDICT for the synthetic dataset.** Comparison of the extrapolation and interpolation calibration of the distributions predicted by EDICT and GRU-ODE-Bayes.

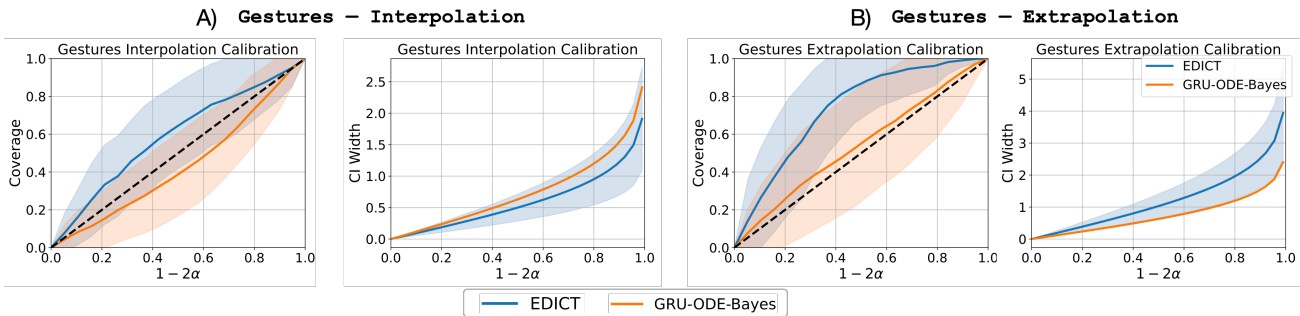

Figure 7: **Calibration of EDICT for the Gestures dataset.** Comparison of the extrapolation and interpolation calibration of the distributions predicted by EDICT and GRU-ODE-Bayes.

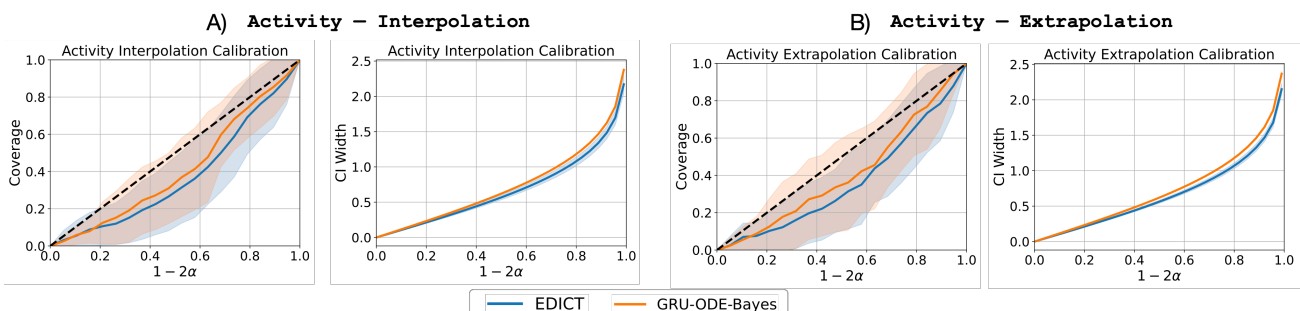

Figure 8: **Calibration of EDICT for the Activity dataset.** Comparison of the extrapolation and interpolation calibration of the distributions predicted by EDICT and GRU-ODE-Bayes.

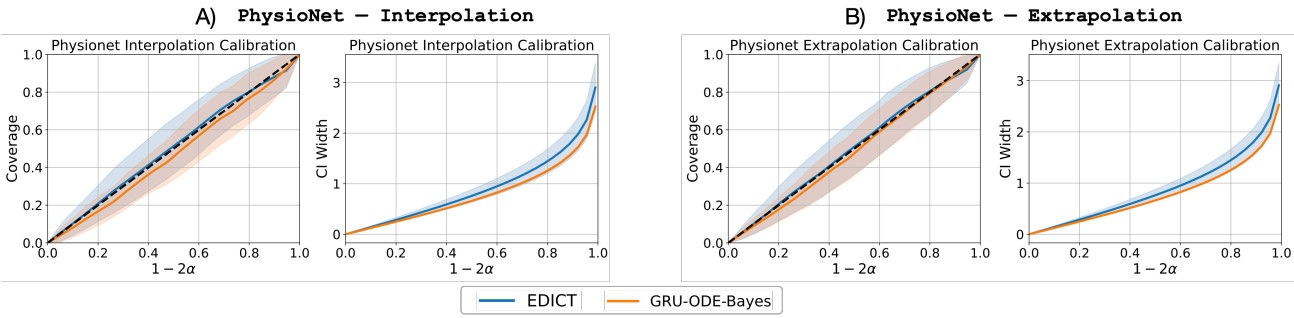

Figure 9: **Calibration of EDICT for the PhysioNet dataset.** Comparison of the extrapolation and interpolation calibration of the distributions predicted by EDICT and GRU-ODE-Bayes.

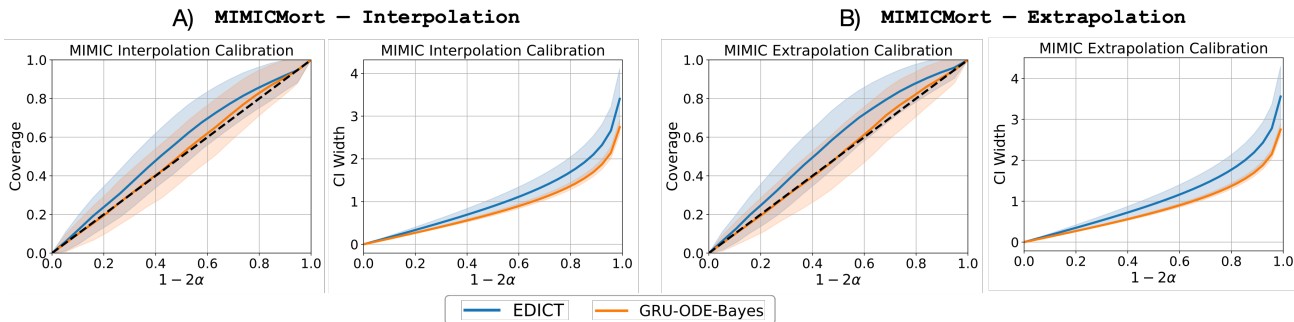

Figure 10: **Calibration of EDICT for the MIMICMort dataset.** Comparison of the extrapolation and interpolation calibration of the distributions predicted by EDICT and GRU-ODE-Bayes.

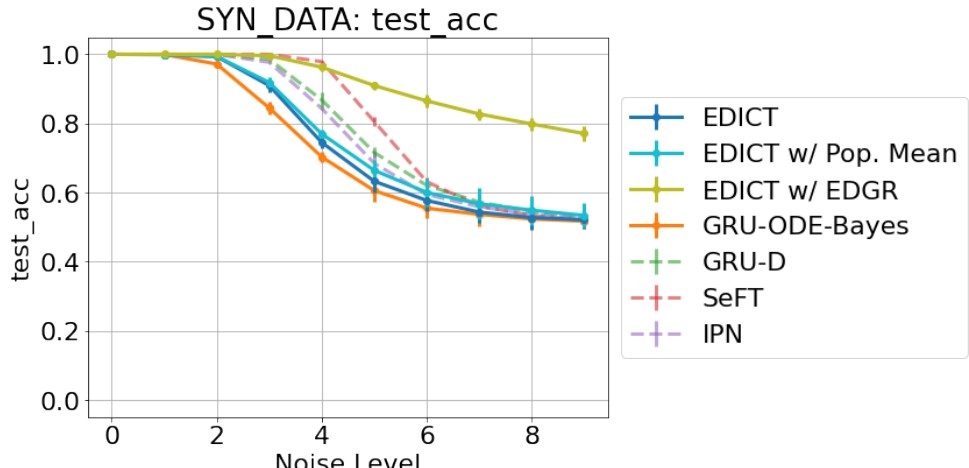

Figure 11: **Classification performance comparison on the Synthetic Dataset**

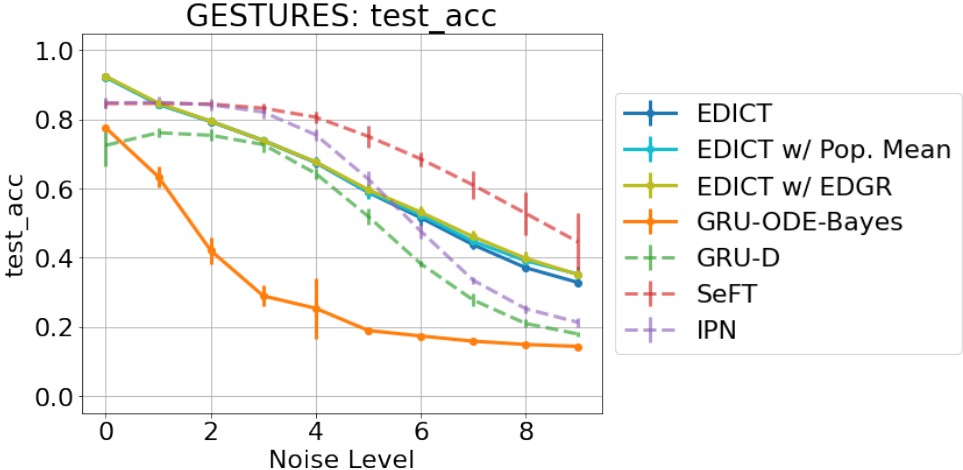

Figure 12: **Classification performance comparison on the Gestures Dataset**

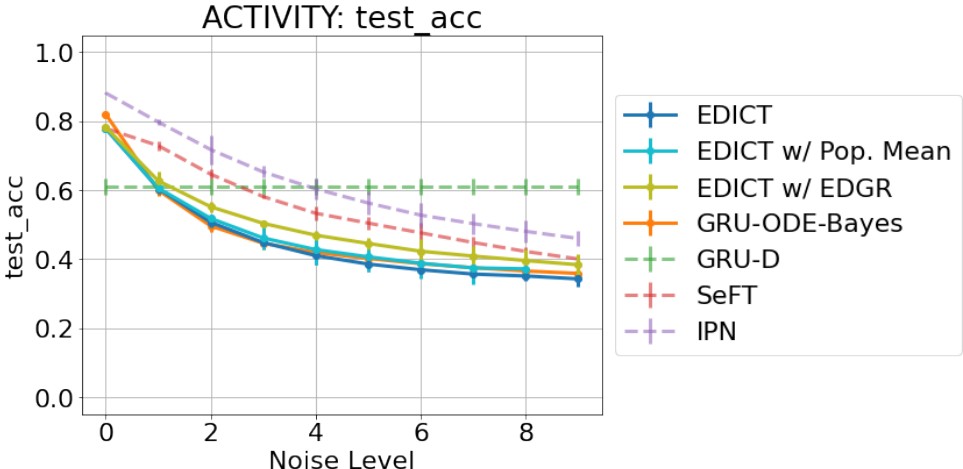

Figure 13: **Classification performance comparison on the Activity Dataset**

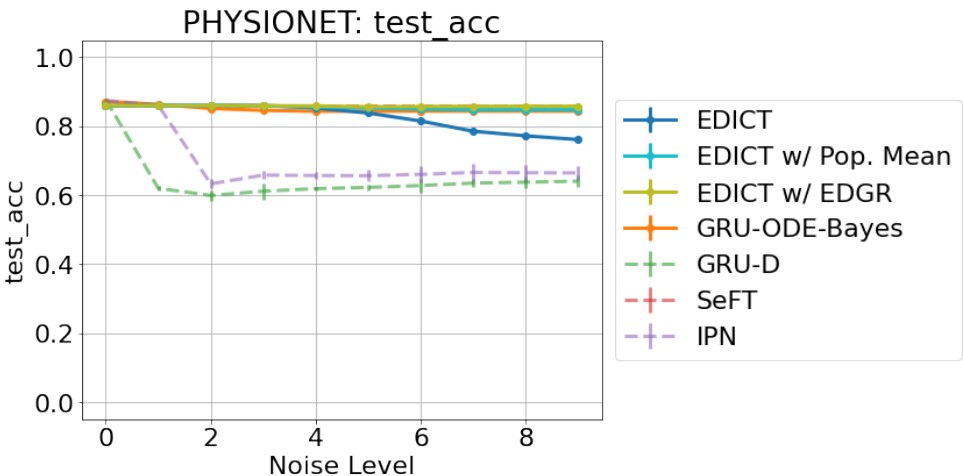

Figure 14: **Classification performance comparison on the Physionet Dataset**

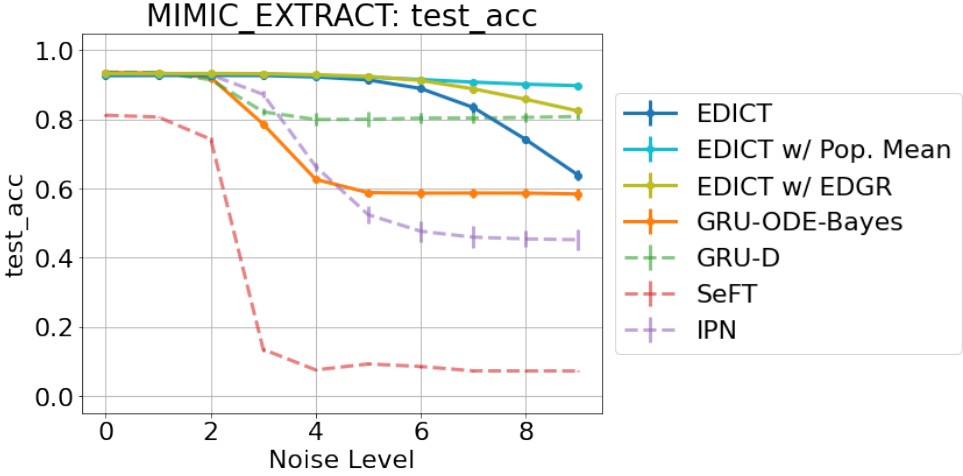

Figure 15: **Classification performance comparison on the MIMIC Dataset**

Table 3: Calibration of EDICT and GRU-ODE-Bayes

| | EDICT | | | | GRU-ODE-Bayes | | | |
| | Interpolation | | Extrapolation | | Interpolation | | Extrapolation | |
| Dataset | MSE | ECE | MSE | ECE | MSE | ECE | MSE | ECE |
|---|---|---|---|---|---|---|---|---|
| Synthetic | $0.124 \pm 0.29$ | $0.083 \pm 0.049$ | $0.009 \pm 0.01$ | $0.129 \pm 0.077$ | $0.769 \pm 0.40$ | $0.068 \pm 0.05$ | $0.696 \pm 0.20$ | $0.093 \pm 0.78$ |
| Gestures | $0.188 \pm 0.02$ | $0.077 \pm 0.01$ | $0.273 \pm 0.02$ | $0.066 \pm 0.04$ | $0.767 \pm 0.48$ | $0.064 \pm 0.03$ | $0.603 \pm 0.12$ | $0.054 \pm 0.05$ |
| Activity | $1.083 \pm 0.78$ | $0.120 \pm 0.07$ | $1.126 \pm 0.72$ | $0.127 \pm 0.07$ | $1.229 \pm 0.92$ | $0.084 \pm 0.05$ | $1.119 \pm 0.76$ | $0.085 \pm 0.05$ |
| PhysioNet | $1.293 \pm 1.26$ | $0.011 \pm 0.01$ | $1.326 \pm 1.46$ | $0.008 \pm 0.07$ | $1.216 \pm 0.72$ | $0.029 \pm 0.01$ | $1.160 \pm 0.76$ | $0.014 \pm 0.01$ |
| MIMICMort | $1.464 \pm 2.70$ | $0.051 \pm 0.03$ | $1.374 \pm 2.68$ | $0.066 \pm 0.04$ | $1.507 \pm 3.48$ | $0.010 \pm 0.01$ | $1.227 \pm 0.94$ | $0.009 \pm 0.01$ |

Table 4: Test-set accuracy (%) of each method on each dataset for varying levels of noise. Confidence intervals are computed as the standard deviation over three random seeds.

| Dataset | Noise Level | EDICT | EDICT w/ EDGR | EDICT w/ Pop. Mean | GRU-ODE-Bayes | GRU-D | IPN | SeFT |
|---|---|---|---|---|---|---|---|---|
| Synthetic | 0 | $100.0 \pm 0.0$ | $100.0 \pm 0.0$ | $100.0 \pm 0.0$ | $100.0 \pm 0.0$ | $100.0 \pm 0.0$ | $100.0 \pm 0.0$ | $100.0 \pm 0.0$ |
| | 1 | $99.9 \pm 0.0$ | $100.0 \pm 0.0$ | $99.9 \pm 0.0$ | $100.0 \pm 0.1$ | $100.0 \pm 0.0$ | $100.0 \pm 0.0$ | $100.0 \pm 0.0$ |
| | 2 | $99.4 \pm 0.2$ | $100.0 \pm 0.0$ | $99.4 \pm 0.2$ | $97.2 \pm 0.5$ | $100.0 \pm 0.0$ | $100.0 \pm 0.0$ | $100.0 \pm 0.0$ |
| | 3 | $91.0 \pm 0.9$ | $99.6 \pm 0.2$ | $91.9 \pm 0.7$ | $84.5 \pm 0.9$ | $98.6 \pm 0.3$ | $97.7 \pm 0.0$ | $100.0 \pm 0.0$ |
| | 4 | $74.5 \pm 0.9$ | $96.3 \pm 0.6$ | $77.0 \pm 0.6$ | $70.3 \pm 0.7$ | $86.8 \pm 1.1$ | $84.3 \pm 0.1$ | $97.9 \pm 0.1$ |
| | 5 | $63.3 \pm 1.5$ | $91.0 \pm 0.6$ | $66.4 \pm 1.5$ | $60.6 \pm 1.7$ | $71.6 \pm 0.8$ | $68.5 \pm 1.6$ | $80.6 \pm 0.7$ |
| | 6 | $57.7 \pm 1.6$ | $86.5 \pm 0.9$ | $60.0 \pm 2.0$ | $55.5 \pm 1.4$ | $62.1 \pm 0.8$ | $59.4 \pm 1.7$ | $63.1 \pm 0.7$ |
| | 7 | $54.4 \pm 1.6$ | $82.7 \pm 0.8$ | $56.8 \pm 2.3$ | $53.7 \pm 1.8$ | $57.2 \pm 1.1$ | $55.9 \pm 1.0$ | $56.1 \pm 0.7$ |
| | 8 | $52.8 \pm 1.8$ | $79.8 \pm 0.9$ | $54.9 \pm 2.1$ | $52.3 \pm 1.3$ | $54.4 \pm 0.8$ | $53.6 \pm 1.3$ | $53.4 \pm 0.6$ |
| | 9 | $52.1 \pm 1.4$ | $77.1 \pm 1.0$ | $53.4 \pm 1.8$ | $51.7 \pm 1.1$ | $53.0 \pm 0.9$ | $52.5 \pm 1.3$ | $52.0 \pm 0.5$ |
| Gestures | 0 | $92.3 \pm 0.3$ | $92.5 \pm 0.2$ | $92.2 \pm 0.3$ | $77.6 \pm 0.4$ | $76.2 \pm 1.4$ | $84.9 \pm 1.6$ | $84.6 \pm 0.7$ |
| | 1 | $84.5 \pm 0.3$ | $84.7 \pm 0.2$ | $84.4 \pm 0.4$ | $63.4 \pm 3.2$ | $75.4 \pm 1.7$ | $84.3 \pm 1.7$ | $84.4 \pm 0.9$ |
| | 2 | $79.4 \pm 0.3$ | $79.5 \pm 0.3$ | $79.5 \pm 0.4$ | $42.1 \pm 3.9$ | $72.7 \pm 2.4$ | $82.1 \pm 1.9$ | $83.3 \pm 1.2$ |
| | 3 | $73.8 \pm 0.6$ | $73.9 \pm 0.8$ | $74.0 \pm 0.9$ | $28.9 \pm 3.1$ | $64.4 \pm 1.5$ | $75.5 \pm 1.8$ | $80.7 \pm 1.7$ |
| | 4 | $67.5 \pm 1.5$ | $67.7 \pm 1.2$ | $67.6 \pm 1.3$ | $25.3 \pm 8.9$ | $51.9 \pm 2.5$ | $62.8 \pm 2.3$ | $75.1 \pm 3.1$ |
| | 5 | $58.9 \pm 1.6$ | $59.7 \pm 1.7$ | $59.0 \pm 1.9$ | $18.9 \pm 0.9$ | $38.3 \pm 0.7$ | $47.7 \pm 2.1$ | $68.6 \pm 2.1$ |
| | 6 | $51.6 \pm 2.4$ | $53.2 \pm 1.7$ | $52.3 \pm 2.1$ | $17.3 \pm 0.7$ | $27.8 \pm 1.8$ | $33.5 \pm 1.0$ | $61.0 \pm 4.0$ |
| | 7 | $43.8 \pm 1.4$ | $46.1 \pm 1.6$ | $44.7 \pm 1.7$ | $15.8 \pm 0.6$ | $20.9 \pm 1.2$ | $25.2 \pm 1.2$ | $52.8 \pm 6.2$ |
| | 8 | $37.1 \pm 1.1$ | $39.8 \pm 1.9$ | $39.1 \pm 1.8$ | $14.9 \pm 0.7$ | $17.9 \pm 0.6$ | $21.3 \pm 1.3$ | $44.5 \pm 8.3$ |
| | 9 | $32.8 \pm 1.1$ | $35.0 \pm 1.6$ | $35.2 \pm 2.0$ | $14.3 \pm 0.5$ | $16.1 \pm 1.1$ | $18.6 \pm 0.8$ | $37.2 \pm 9.4$ |
| Activity | 0 | $77.9 \pm 0.2$ | $78.2 \pm 0.3$ | $77.9 \pm 0.1$ | $82.0 \pm 0.1$ | $61.1 \pm 1.1$ | $88.2 \pm 0.0$ | $77.9 \pm 0.1$ |
| | 1 | $60.5 \pm 0.7$ | $62.7 \pm 1.3$ | $60.6 \pm 0.5$ | $60.0 \pm 0.9$ | $61.1 \pm 1.1$ | $79.8 \pm 0.4$ | $72.9 \pm 0.7$ |
| | 2 | $50.7 \pm 0.8$ | $55.2 \pm 0.8$ | $51.8 \pm 0.4$ | $49.6 \pm 0.8$ | $61.1 \pm 1.1$ | $71.8 \pm 2.1$ | $64.6 \pm 0.7$ |
| | 3 | $44.8 \pm 0.8$ | $50.4 \pm 0.5$ | $46.1 \pm 1.6$ | $44.6 \pm 0.8$ | $61.1 \pm 1.1$ | $65.3 \pm 0.8$ | $58.1 \pm 0.2$ |
| | 4 | $41.0 \pm 1.4$ | $46.9 \pm 0.7$ | $42.8 \pm 2.2$ | $41.9 \pm 0.4$ | $61.1 \pm 1.1$ | $60.4 \pm 1.5$ | $53.3 \pm 1.0$ |
| | 5 | $38.6 \pm 1.1$ | $44.5 \pm 0.8$ | $40.7 \pm 2.1$ | $40.2 \pm 0.6$ | $61.1 \pm 1.1$ | $56.3 \pm 1.7$ | $50.4 \pm 0.9$ |
| | 6 | $36.9 \pm 0.4$ | $42.3 \pm 1.7$ | $38.8 \pm 2.3$ | $38.7 \pm 1.0$ | $61.1 \pm 1.1$ | $52.8 \pm 1.8$ | $47.6 \pm 1.5$ |
| | 7 | $35.7 \pm 0.6$ | $40.9 \pm 1.5$ | $37.5 \pm 2.4$ | $37.5 \pm 0.7$ | $61.1 \pm 1.1$ | $50.3 \pm 1.6$ | $44.8 \pm 0.9$ |
| | 8 | $35.1 \pm 0.7$ | $39.6 \pm 1.6$ | $37.2 \pm 0.7$ | $36.6 \pm 0.9$ | $61.1 \pm 1.1$ | $48.1 \pm 1.6$ | $42.2 \pm 0.6$ |
| | 9 | $34.3 \pm 1.1$ | $38.4 \pm 1.6$ | $36.8 \pm 0.0$ | $35.9 \pm 0.8$ | $61.1 \pm 1.1$ | $46.0 \pm 1.2$ | $40.0 \pm 0.5$ |
| PhysioNet | 0 | $85.9 \pm 0.0$ | $86.0 \pm 0.1$ | $86.0 \pm 0.1$ | $87.1 \pm 0.0$ | $87.7 \pm 0.2$ | $87.5 \pm 0.1$ | $86.0 \pm 0.0$ |
| | 1 | $86.0 \pm 0.1$ | $86.0 \pm 0.1$ | $86.0 \pm 0.1$ | $86.3 \pm 0.3$ | $62.0 \pm 0.4$ | $86.0 \pm 0.3$ | $86.0 \pm 0.0$ |
| | 2 | $86.0 \pm 0.1$ | $86.0 \pm 0.1$ | $86.0 \pm 0.1$ | $85.1 \pm 0.2$ | $59.9 \pm 0.7$ | $63.3 \pm 0.7$ | $86.0 \pm 0.0$ |
| | 3 | $86.0 \pm 0.2$ | $86.0 \pm 0.0$ | $85.9 \pm 0.1$ | $84.6 \pm 0.1$ | $61.1 \pm 1.1$ | $65.8 \pm 0.6$ | $86.0 \pm 0.0$ |
| | 4 | $85.3 \pm 0.2$ | $85.9 \pm 0.1$ | $85.6 \pm 0.2$ | $84.3 \pm 0.3$ | $61.9 \pm 0.3$ | $65.7 \pm 0.5$ | $86.0 \pm 0.0$ |
| | 5 | $83.8 \pm 0.1$ | $85.6 \pm 0.2$ | $85.3 \pm 0.1$ | $84.3 \pm 0.2$ | $62.2 \pm 0.4$ | $65.6 \pm 0.8$ | $86.0 \pm 0.0$ |
| | 6 | $81.5 \pm 0.4$ | $85.7 \pm 0.1$ | $85.0 \pm 0.3$ | $84.4 \pm 0.1$ | $62.8 \pm 1.0$ | $66.0 \pm 1.2$ | $86.0 \pm 0.0$ |
| | 7 | $78.5 \pm 0.1$ | $85.7 \pm 0.1$ | $84.8 \pm 0.2$ | $84.4 \pm 0.1$ | $63.5 \pm 0.6$ | $66.6 \pm 1.2$ | $86.0 \pm 0.0$ |
| | 8 | $77.2 \pm 0.2$ | $85.8 \pm 0.1$ | $84.8 \pm 0.2$ | $84.4 \pm 0.1$ | $63.8 \pm 0.9$ | $66.5 \pm 1.2$ | $86.0 \pm 0.0$ |
| | 9 | $76.1 \pm 0.1$ | $85.8 \pm 0.2$ | $84.8 \pm 0.2$ | $84.4 \pm 0.1$ | $64.0 \pm 0.9$ | $66.5 \pm 1.0$ | $86.0 \pm 0.0$ |
| MIMICMort | 0 | $92.7 \pm 0.1$ | $93.3 \pm 0.1$ | $93.1 \pm 0.0$ | $93.6 \pm 0.1$ | $93.7 \pm 0.1$ | $93.4 \pm 0.1$ | $81.2 \pm 0.2$ |
| | 1 | $92.7 \pm 0.1$ | $93.3 \pm 0.1$ | $93.2 \pm 0.1$ | $93.4 \pm 0.0$ | $93.5 \pm 0.2$ | $93.3 \pm 0.2$ | $80.7 \pm 0.1$ |
| | 2 | $92.8 \pm 0.1$ | $93.3 \pm 0.1$ | $93.2 \pm 0.1$ | $92.1 \pm 0.1$ | $91.5 \pm 0.4$ | $92.9 \pm 0.2$ | $74.2 \pm 0.4$ |
| | 3 | $92.7 \pm 0.2$ | $93.2 \pm 0.2$ | $93.1 \pm 0.2$ | $78.6 \pm 0.4$ | $82.2 \pm 0.5$ | $87.3 \pm 0.5$ | $13.4 \pm 0.3$ |
| | 4 | $92.3 \pm 0.3$ | $92.9 \pm 0.2$ | $92.7 \pm 0.1$ | $62.7 \pm 0.1$ | $80.0 \pm 0.9$ | $66.4 \pm 0.6$ | $7.5 \pm 0.0$ |
| | 5 | $91.4 \pm 0.3$ | $92.5 \pm 0.4$ | $92.2 \pm 0.3$ | $58.8 \pm 0.4$ | $80.1 \pm 1.1$ | $52.4 \pm 1.3$ | $9.2 \pm 0.0$ |
| | 6 | $89.0 \pm 0.1$ | $91.3 \pm 0.3$ | $91.6 \pm 0.3$ | $58.7 \pm 0.6$ | $80.4 \pm 0.8$ | $47.7 \pm 1.5$ | $8.5 \pm 0.2$ |
| | 7 | $83.5 \pm 0.7$ | $88.9 \pm 0.3$ | $90.8 \pm 0.1$ | $58.7 \pm 0.7$ | $80.4 \pm 0.7$ | $45.9 \pm 1.6$ | $7.2 \pm 0.0$ |
| | 8 | $74.2 \pm 0.4$ | $85.8 \pm 0.2$ | $90.2 \pm 0.2$ | $58.7 \pm 0.7$ | $80.6 \pm 0.7$ | $45.4 \pm 1.2$ | $7.2 \pm 0.0$ |
| | 9 | $63.9 \pm 0.7$ | $82.5 \pm 0.0$ | $89.8 \pm 0.2$ | $58.4 \pm 0.8$ | $80.8 \pm 0.5$ | $45.2 \pm 1.5$ | $7.2 \pm 0.0$ |

Table 5: Test-set AUROC (%) of each method on each binary classification dataset for varying levels of noise. Confidence intervals are computed as the standard deviation over three random seeds.

| Dataset | Noise Level | EDICT | EDICT w/ EDGR | EDICT w/ Pop. Mean | GRU-ODE-Bayes | GRU-D | IPN | SeFT |
|---|---|---|---|---|---|---|---|---|
| Synthetic | 0 | $100.0 \pm 0.0$ | $100.0 \pm 0.0$ | $100.0 \pm 0.0$ | $100.0 \pm 0.0$ | $100.0 \pm 0.0$ | $100.0 \pm 0.0$ | $100.0 \pm 0.0$ |
| | 1 | $100.0 \pm 0.0$ | $100.0 \pm 0.0$ | $100.0 \pm 0.0$ | $100.0 \pm 0.0$ | $100.0 \pm 0.0$ | $100.0 \pm 0.0$ | $100.0 \pm 0.0$ |
| | 2 | $100.0 \pm 0.0$ | $100.0 \pm 0.0$ | $100.0 \pm 0.0$ | $99.7 \pm 0.1$ | $100.0 \pm 0.0$ | $100.0 \pm 0.0$ | $100.0 \pm 0.0$ |
| | 3 | $97.0 \pm 0.5$ | $100.0 \pm 0.0$ | $97.8 \pm 0.3$ | $93.6 \pm 0.4$ | $99.8 \pm 0.2$ | $99.7 \pm 0.0$ | $100.0 \pm 0.0$ |
| | 4 | $83.4 \pm 1.1$ | $99.6 \pm 0.2$ | $86.9 \pm 0.9$ | $79.3 \pm 0.9$ | $93.0 \pm 0.8$ | $92.1 \pm 0.1$ | $99.7 \pm 0.0$ |
| | 5 | $69.4 \pm 1.1$ | $97.4 \pm 0.4$ | $74.3 \pm 1.2$ | $67.2 \pm 1.2$ | $77.9 \pm 0.3$ | $75.7 \pm 0.6$ | $86.2 \pm 0.6$ |
| | 6 | $61.0 \pm 1.0$ | $94.3 \pm 0.7$ | $65.6 \pm 1.1$ | $59.8 \pm 1.5$ | $67.0 \pm 0.7$ | $65.1 \pm 0.8$ | $65.4 \pm 0.7$ |
| | 7 | $56.5 \pm 0.9$ | $90.8 \pm 0.8$ | $60.5 \pm 1.3$ | $55.8 \pm 1.5$ | $60.7 \pm 0.9$ | $59.5 \pm 0.8$ | $56.9 \pm 0.9$ |
| | 8 | $54.3 \pm 0.8$ | $87.6 \pm 1.0$ | $57.3 \pm 1.2$ | $53.7 \pm 1.4$ | $57.3 \pm 0.9$ | $56.6 \pm 0.9$ | $53.7 \pm 0.8$ |
| | 9 | $53.0 \pm 0.7$ | $84.7 \pm 1.3$ | $55.3 \pm 1.3$ | $52.5 \pm 1.4$ | $55.3 \pm 0.8$ | $54.6 \pm 0.8$ | $51.9 \pm 0.7$ |
| PhysioNet | 0 | $69.9 \pm 0.1$ | $69.7 \pm 0.1$ | $69.7 \pm 0.1$ | $79.2 \pm 0.2$ | $83.0 \pm 0.1$ | $82.7 \pm 0.1$ | $68.3 \pm 0.1$ |
| | 1 | $69.7 \pm 0.3$ | $69.6 \pm 0.2$ | $69.5 \pm 0.3$ | $69.4 \pm 2.5$ | $55.0 \pm 0.8$ | $79.7 \pm 0.4$ | $66.1 \pm 0.5$ |
| | 2 | $69.1 \pm 0.6$ | $68.9 \pm 0.4$ | $68.9 \pm 0.5$ | $53.5 \pm 3.3$ | $50.6 \pm 1.2$ | $57.5 \pm 1.2$ | $52.4 \pm 2.4$ |
| | 3 | $67.4 \pm 0.9$ | $67.2 \pm 0.7$ | $67.2 \pm 0.7$ | $52.0 \pm 3.5$ | $50.6 \pm 1.5$ | $51.1 \pm 0.9$ | $44.0 \pm 2.3$ |
| | 4 | $64.2 \pm 1.3$ | $64.3 \pm 1.0$ | $64.5 \pm 0.9$ | $51.5 \pm 3.5$ | $50.4 \pm 1.3$ | $50.9 \pm 0.9$ | $46.1 \pm 0.3$ |
| | 5 | $60.4 \pm 1.7$ | $61.3 \pm 2.1$ | $61.6 \pm 1.3$ | $51.5 \pm 3.4$ | $50.5 \pm 1.0$ | $50.8 \pm 1.0$ | $45.6 \pm 0.5$ |
| | 6 | $57.7 \pm 2.0$ | $58.8 \pm 3.1$ | $59.2 \pm 2.0$ | $51.5 \pm 3.5$ | $50.7 \pm 1.0$ | $51.2 \pm 1.3$ | $49.7 \pm 0.5$ |
| | 7 | $56.5 \pm 2.0$ | $57.5 \pm 3.0$ | $57.9 \pm 2.4$ | $51.4 \pm 3.5$ | $50.5 \pm 0.7$ | $51.4 \pm 1.1$ | $49.9 \pm 0.0$ |
| | 8 | $55.8 \pm 1.9$ | $56.7 \pm 2.7$ | $57.3 \pm 2.4$ | $51.5 \pm 3.5$ | $50.7 \pm 0.7$ | $51.3 \pm 1.3$ | $49.7 \pm 0.1$ |
| | 9 | $55.6 \pm 1.9$ | $55.2 \pm 2.5$ | $57.2 \pm 2.3$ | $51.5 \pm 3.5$ | $50.7 \pm 0.8$ | $51.3 \pm 1.1$ | $49.8 \pm 0.0$ |
| MIMICMort | 0 | $83.9 \pm 0.1$ | $83.2 \pm 0.0$ | $83.1 \pm 0.1$ | $88.5 \pm 0.1$ | $89.6 \pm 0.1$ | $86.7 \pm 0.0$ | $74.4 \pm 0.1$ |
| | 1 | $83.9 \pm 0.1$ | $83.2 \pm 0.1$ | $83.2 \pm 0.2$ | $88.2 \pm 0.2$ | $89.2 \pm 0.2$ | $86.6 \pm 0.1$ | $74.4 \pm 0.0$ |
| | 2 | $83.8 \pm 0.2$ | $83.1 \pm 0.1$ | $83.2 \pm 0.3$ | $83.7 \pm 0.3$ | $83.1 \pm 0.8$ | $85.7 \pm 0.5$ | $72.5 \pm 0.1$ |
| | 3 | $83.6 \pm 0.3$ | $82.8 \pm 0.2$ | $83.0 \pm 0.4$ | $69.0 \pm 0.4$ | $63.7 \pm 1.8$ | $77.8 \pm 0.7$ | $58.0 \pm 0.4$ |
| | 4 | $82.7 \pm 0.5$ | $82.0 \pm 0.3$ | $82.2 \pm 0.4$ | $56.2 \pm 0.6$ | $55.0 \pm 1.7$ | $64.7 \pm 1.4$ | $51.4 \pm 0.6$ |
| | 5 | $80.6 \pm 0.5$ | $80.0 \pm 0.4$ | $80.4 \pm 0.4$ | $52.3 \pm 0.7$ | $52.3 \pm 1.2$ | $56.8 \pm 1.7$ | $49.4 \pm 0.5$ |
| | 6 | $76.9 \pm 0.3$ | $76.4 \pm 0.4$ | $77.7 \pm 0.4$ | $51.4 \pm 1.0$ | $51.5 \pm 0.8$ | $53.4 \pm 2.0$ | $50.4 \pm 0.6$ |
| | 7 | $71.9 \pm 0.5$ | $71.5 \pm 0.7$ | $74.9 \pm 0.6$ | $51.0 \pm 1.3$ | $51.2 \pm 0.4$ | $51.9 \pm 1.9$ | $50.2 \pm 0.2$ |
| | 8 | $66.7 \pm 0.9$ | $66.8 \pm 1.1$ | $72.0 \pm 0.6$ | $50.9 \pm 1.4$ | $51.0 \pm 0.7$ | $51.2 \pm 1.8$ | $50.0 \pm 0.0$ |
| | 9 | $61.5 \pm 0.3$ | $63.5 \pm 1.3$ | $70.4 \pm 0.6$ | $50.6 \pm 1.2$ | $51.0 \pm 0.8$ | $51.0 \pm 1.8$ | $49.9 \pm 0.0$ |

