# OpenReview forum: "Continuous Time Evidential Distributions for Irregular Time Series"
_ICML.cc/2023/Workshop/IMLH — IMLH 2023 Poster_

### Official Review · Reviewer_sKaw · 2023-06-11
**Good paper**

**Rating:** 8
**Confidence:** 4

**Review:**

Paper type: 8-page long paper.

## Summary

This paper proposes a method called Evidential Distributions In Continuous Time (EDICT) that can learn the evidential distribution of a hidden representation \Phi from sporadically-observed time series in continuous time. The EDICT model assumes that the hidden representation \Phi is a multivariate normal distribution. It first formulates the past observations in a continuous-time recurrent latent representation h(t) using a neural ordinary differential equation (ODE) framework. Once the h(t) is constructed, the hidden representation \Phi of the dataset can be learned from a Bayes-based evidential learning framework that maximizes the likelihood of the observation at time t. All the learning of the parameters in these steps are built in neural networks.

This EDICT model provides calibrated, and uncertainty-aware estimations over time without sampling during inference Moreover, the paper created an algorithm called Evidential Distribution Guided Reweighting (EDGR) that utilize the uncertainty estimation of EDICT to identify the noise and outliers in the downstream prediction task such as classification. The EDGR has been shown to effectively improve the robustness of the classification result at different noise levels. Experiments on both synthetic and multiple real-world datasets have shown that the proposed method EDICT has shown competitive performance on challenging time series classification tasks in comparison to existing state-of-the-art methods such as GRU-ODE-Bayes.

## Strength

This paper presents a novel idea that formulates the learning of a continuous time representation in the evidential learning framework. This avoids the sampling during inference time and can produce calibrated probability with uncertainty estimation.

The paper has conducted a comprehensive evaluation to validate the performance of the proposed method (EDICT and EDGR). The baseline setup (i.e. GRU-ODE-Bayes and other NN methods) is appropriate, which demonstrates the competitive performance of the EDICT and EDGR. The evaluation results show the strong capability of the proposed method in the uncertainty estimation, calibration quality (with some questionable results, see Weakness section), performance in the downstream tasks, and prediction over noisy observations. Evaluation on both real world and synthetic datasets demonstrates the applicability of the method, which makes me believe that the proposed solution could have a substantial impact on the research community.

I also commend the authors analysis on the limitations of their proposed method as well as its potential negative societal impact from the misuse of the algorithm. This shows not only their strong technical knowledge but also their thoughtful consideration for the general public.

Overall this paper is well-structured and easy to follow. The methodology is clearly described, and the logic flows smoothly. I especially appreciate the occasional reminder of the related sections of a given terminology.

## Weakness

The evaluation of the calibration quality of the EDICT primarily focuses on the MSE metric instead of the Expected Calibration Error (ECE) metric. Table 2 shows that the baseline GRU-ODE-Bayes has lower ECE than EDICT, which indicates that GRU-ODE-Bayes has better calibration quality for the uncertainty estimation. No analysis is given from this perspective, but given that the ECE value between the two methods are close, I think the claim that EDICT produces a well-calibrated model uncertainties still holds.

Further investigation is needed to explain why EDICT with Population Mean has a better AUROC than EDICT with EDGR in the MIMICMort dataset.

I am skeptical of the claim that the EDICT model is trained in an unsupervised manner. After all, how to generate the negative log-likelihood loss, KL loss and regulation loss without supervision when all of them are based on the observed evidence? Isn’t evidence the same as the labels? The usage of the evidence (which I think is label) is also shown in the Appendix B.

A more detailed diagram of the EDICT architecture, including its components such as f_ODE, f_Baves, and f_NIW, would be helpful to understand the network architecture.

Minor writing suggestions: There is a typo at line 256. The utility of EDGR is shown in Section 5.3, not 5.2.

## Rating and Justification

This paper proposes a novel idea to learn from irregular time series data in an evidential learning framework. The paper provides comprehensive analysis and explanation in both the main paper and supplementary materials, demonstrating the strong capability of its proposed model in the estimation of the hidden representation in the continuous time. It also discusses the limitations of the model and the potential damage that could be caused by its misuse. I therefore recommend that this paper be accepted for publication.

---

### Official Review · Reviewer_3xrE · 2023-06-19
**Well-fitted method, comprehensive experiments.**

**Rating:** 9
**Confidence:** 3

**Review:**

Considering the inherent challenges of time series sampling, which often presents noisy and sporadic data, this paper presents a method for learning a latent representation from these irregular time series. As a result,  by using neural ODE, this work enables calibrated, temporally correlated uncertainty estimates.
Pros:
1. The application of Ordinary Differential Equations (ODE) is aptly suited to this context as ODE allows for temporal interpolation. This can yield a regular, continuous time series output – in this case, uncertainty estimation.
2. The paper is clearly written, with demonstrations and figure illustrations providing informative visuals.
3. The authors validate their methodology through experiments conducted on a variety of temporal series datasets, including both synthetic and real-world examples. This underscores the general applicability of the proposed framework.
4. The authors also offer a commendable discussion on the limitations of their work.
Cons: At this stage, I find no apparent shortcomings in the study.

---

### Meta-Review · Area_Chair_vAUM · 2023-06-18

**Recommendation:** Accept (Poster)
**Confidence:** 4

**Metareview:**

The proposed method in this paper is EDICT (Evidential Distributions in Continuous Time), which can learn the evidential distribution of a hidden representation, Phi, from sporadically-observed time series in continuous time.

The paper has been praised by reviewers for being well-written, well-motivated, and for having thorough evaluations and interesting methods. There were some concerns raised by Reviewer sKaw, which the authors should take into account when preparing the camera-ready version.

---

### Decision · Program_Chairs · 2023-06-20

Accept (Poster)